# Incidence and Spatial Mapping of Tuberculosis and Multidrug-Resistant Tuberculosis in Libreville, Republic of Gabon, in 2022

**DOI:** 10.3390/tropicalmed11010008

**Published:** 2025-12-27

**Authors:** Casimir Manzengo, Nlandu Roger Ngatu, Stredice Manguinga-Guitouka, Fleur Lignenguet, Ghislaine Nkone-Asseko, Marie Nsimba-Miezi, Nobuyuki Miyatake, Jose Lami-Nzunzu, Tomohiro Hirao

**Affiliations:** 1World Health Organization (WHO)-Burkina Faso, Ouagadougou BP 7019, Burkina Faso; 2Department of Public Health, Kagawa University Faculty of Medicine, Kagawa 761-0701, Japan; 3Programme National Tuberculose (PNT), Ministère de la Santé, Libreville BP 50, Gabon; 4World Health Organization (WHO)-Gabon, Libreville BP 820, Gabon; 5Department of Biopharmaceutical and Food Sciences, Faculty of Pharmaceutical Sciences, University of Kinshasa, Kinshasa P.O. Box 212, Congo; 6Department of Hygiene, Kagawa University Faculty of Medicine, Kagawa 761-0701, Japan; 7Department of Medicinal Chemistry and Pharmacognosy, Faculty of Pharmaceutical Sciences, University of Kinshasa, Kinshasa P.O. Box 212, Congo

**Keywords:** infection, Libreville, multi-drug-resistant tuberculosis, tuberculosis

## Abstract

**Background:** Tuberculosis (TB) remains a major global health problem, and the WHO central Africa region continues to bear the heaviest disease burden. Gabon is one of the high-TB-burden countries in the world; however, its national TB program performance remains weak despite financial support from international health agencies. Identifying and mapping high-TB- and multi-drug-resistant-TB (MDR-TB)-burden areas for targeted public health interventions was the objective of this study. **Methods:** A region-wide mixed method study was carried out, comprising ecological design and a desk review, with the use of medical records from TB diagnosis and care units in 12 health facilities located across the capital Libreville, Republic of Gabon, from 1 January through December 2022. Libreville is the region that bears the heaviest TB burden in Gabon. With the collaboration of the Agency for Space Studies and Observations (AGEOS, Gabon), collected data were transferred to and analyzed using QGIS software in order to develop satellite images. **Results:** In the Libreville health region, there were 4560 cases diagnosed in 2022, representing 77.9% of all cases in the country, with an annual incidence of 509 per 100,000. Spatial mapping of incident cases by county of residence showed that a large majority of the TB cases diagnosed at the CHUL care center in 2022 were from Nzeng-Ayong (range: 36–50 cases) and Owendo (26–35 cases), whereas higher TB incidence at the Nkembo care center was from Nzeng-Ayong (range: 356–455 cases) and Owendo (256–355 cases), followed by Nkembo, Akebe ville, Akebe Baraka, Akebe Plaine/plateau, Angondje, Angondje village, Charbonnages, Bikele, Pk11, Pk12, Pk9, Mindoube I, Mindoube II (66–255 cases), Sotega, and Nkok (46–65 cases). Other counties accounted for less than 45 TB cases. Considering MDR-TB cases, higher incidence was observed in Pk9 county, which accounted for six cases (14.6%), followed by Owendo, accounting for four (9.7%). **Discussion:** Findings suggest that Nzeng-Ayong and Owendo are high-TB-burden counties in Libreville, whereas Pk9 and Owendo counties are counties categorized as high-MDR-TB-incidence areas. They should be subject to targeted to public health interventions to enhance TB control in Libreville.

## 1. Introduction

Tuberculosis (TB), a preventable infectious disease caused by *M. tuberculosis*, remains a major health problem in Sub-Saharan Africa (SSA) [1]. A recent report from the World Health Organization (WHO), which covers more than 99% of the world’s population, stipulated that TB ranked second among leading causes of death from a single infectious agent in the year 2022 [2]. Although some regions of the world have achieved substantial reduction in TB burden, the disease morbidity has increased in developing countries. According to the WHO, TB mortality was 1.6 million in 2022, whereas its prevalence was 10.6 million worldwide, with an increase of over 4% compared to the previous year [3,4]. Additionally, TB remains a serious public health issue in the SSA due to the HIV pandemic, poverty, the movement of displaced people, and the emergence of drug-resistant *M. tuberculosis* strains. Multidrug-resistant TB (MDR-TB) is a life-threatening form of *M. tuberculosis* that may affect other organs, such as the central nervous system, which is associated with neurological complications and high mortality [2,5].

In the Republic of Gabon, TB remains a major public health issue, with Libreville, the capital town, bearing the heaviest burden in the country; it has been identified as an important cause of morbidity and mortality in this country. Despite international support by the Global Fund, the Gabonese National Tuberculosis Program (NTP) has been facing various challenges, including low TB detection and low treatment success rates. In 2021, the TB detection rate was 42%, and the treatment success rate was 57%. Moreover, the HIV-positive rate among TB patients was 29%; however, according to the 2022 NTP annual report, only one third of TB patients have undergone HIV testing [6].

To strengthen the TB program, the WHO Africa has planned to provide technical support for the external review of the Gabonese national TB program to address inherent weaknesses for better outcomes. Prior to implementing such as an intervention, a thorough assessment of high-TB-burden areas and the effectiveness of the national TB program is of utmost importance in Gabon. Detecting emerging high-disease-burden areas can trigger rapid response mechanisms. This helps also us understand transmission dynamics, as tuberculosis is often spread within household or communities, leading to heterogeneous spatial patterns. By mapping TB, health authorities and policymakers can optimize the placement of diagnostic centers, treatment facilities, and outreach programs.

Spatial tools allow real-time surveillance and trend analysis, helping programs assess the impact of interventions over time. Spatial data can reveal inequities in healthcare access, such as underserved rural or peri-urban areas, guiding efforts to improve universal health coverage [7]. The present study aimed to determine the incidence and spatial distribution of TB and multi-drug-resistant TB (MDR-TB) cases in Libreville. This will improve new TB detection among the contacts of confirmed cases thanks to the identification of high-burden areas and inform targeted public health interventions.

## 2. Methods

### 2.1. Design and Settings

This was a mixed method study that comprised an ecological design and a desk review conducted in Libreville, the capital town of the Republic of Gabon. The estimated population was 2,344,720 as of 2022. In the Gabonese health system, the country is divided into 10 health regions, including Libreville (LBV), Ouest, Sud-Est, Centre, Centre-Sud, Sud, Est, Centre-Est, Maritime, and Nord region. Libreville is the most populous province, accounting of 44% (1,029,977 inhabitants) of the entire country population.

Libreville is known as the health region with highest TB burden in Gabon. This is the region where almost 78% of all Gabonese tuberculosis cases in 2022 were detected. Thus, it was selected as the study site. The study covers all new susceptible TB and multi-drug-resistant TB (MDR-TB) cases diagnosed in Libreville from 1 January through December 2022. The Libreville health region has 12 TB diagnosis and care centers located in hospitals and clinics, namely Nkembo, Hiaobo, HIAA, CHUL, CTA CHUL, PCL, CTA Nkembo, Chumje, PLIST, Nvolane, CS Nzeng-Ayong, and Clinique Privee Pasteur. Almost 98% of tuberculosis cases recorded in Libreville in 2022 were detected in the Nkembo and CHUL health centers. Thus, they were selected as the study sites.

### 2.2. Data Sampling

This was a two-stage stratified sampling study. The first stage was the health region, and the health facility was the second one. Considering burden, 78% of all tuberculosis cases were in Libreville, the capital city of Gabon, and 22% in other regions. In Libreville, 98% were diagnosed in two health facilities, which were CDT Nkembo and CHUL. Only 2% were from the 10 others. So, Libreville was considered in the study at the first stage. Then, CDT Nkembo and CHUL were considered sufficient to provide the spatial distribution of the tuberculosis in Gabon.

### 2.3. Data Collection

For data collection, 10 M&E staff from the NTP were briefed on tools used to obtain data from routine records that were the Tb registers and patient treatment file. The tool collected the administrative data, name of the CDT, date of patient registration, CDT register number, sex, age, neighborhood of residence, type of tuberculosis (susceptible, multi-resistant), type of patient, and HIV status (positive, negative, unknown). The patient address was the key information to consider. In the case of a missing address, the record was not considered.

Collected data were scanned, then transcribed in a prepared excel sheet to create the study database, which is available at the Tuberculosis National Programme, Monitoring and Evaluation Department. Only the 2022 period data coming from the Nkembo and CHUL health centers were included in the study.

### 2.4. Data Analysis

Collected data were analyzed using the QGIS (The Ingenuity Centre, Global Geo-Intelligence Solutions Ltd., University of Nottingham, Nottingham, UK), with the technical support of the Gabonese Agency for Space Studies and Observations (AGEOS). QGIS is a geographic information system enabling users to analyze and visualize spatial data. It is an essential tool for addressing health, societal, and environmental challenges and opportunities by transforming aerial data into visual maps [8,9].

The ARCGIS 10.7 software was used, and the data are those of the neighborhood boundaries of Libreville (Source: Ministry of the Interior; Format: shapefile). For neighborhoods that do not yet have recognized boundaries, a buffer zone was used to locate them. The color gradient, for its part, differentiates each neighborhood according to the number of cases observed.

The product was the mapping of TB cases in Libreville, showing the high-burden areas, which can be targeted for specific public health interventions. Satellite images were developed with the use of a geographic information model.

### 2.5. Ethical Considerations

The study was conducted in conformity with internationally approved ethical guidelines. Ethical review and approval were waived due to the use of anonymous epidemiological data and the absence of direct involvement of the patients in the study. All the study data were from TB monitoring and surveillance in Gabon and did not include personal information of the patients. For those reasons, the provision of patient’s informed consent was not applicable.

The study data consisted of medical records from TB treatment centers located at selected health facilities. Ten M&E staff from the NTP were briefed on data collection. Afterwards, collected data were scanned ans then transcribed in prepared excel sheet to create the study dataset, which is available at the Tuberculosis National Programme, Monitoring and Evaluation Department. Only the 2022 data from the Nkembo and CHUL health centers were included in the study for reasons provided above.

## 3. Results

### 3.1. Study Population

Overall, 4466 cases were recorded in both health facilities in 2022, 4309 in CDT Nkembo and 157 in CDT CHUL. About 1102 of participants were female and 3364 male, a sex ratio of 0.25. This is shown in the Table 1 below.

The average age was 35 years [34.23–36.25], with 95% CIs from 1 to 92 years. Overall, 80% of recorded tuberculosis patients are between 20 (percentile 10) and 55 years old (percentile 90). The Table 2 is showing that 6% are under 15 years, 15% are 50 years and old. The remaining patients are in the life-active age, between 15 and 49 years.

From this number, the Table 3 is showing that 3266 were bacteriologically confirmed (73%), 764 were clinically diagnosed (17%), and 436 were extrapulmonary tuberculosis (10%). The new cases and relapses represented 4425 cases, and multi-drug-resistant tuberculosis represented 41 cases.

### 3.2. Incidence of Tuberculosis in the Republic of Gabon and Libreville in the Year 2022

Figure 1 shows nation-wide distribution of TB cases in Libreville in the year 2022. At national level, there were 5850 TB cases diagnosed for an annual incidence of 509 per 100,000. In the Libreville–Owendo health region, there were 4560 cases diagnosed in 2022, representing 77.9% of all cases. The remaining 290 TB cases (22.1%) were diagnosed in the other Gabonese health regions (Figure 1).

In Libreville, a significantly higher number of newly diagnosed TB cases was observed at the Nkembo diagnostic and care center, with 4309 out of 4560 cases (94.5%); followed by CHUL, with 157 cases (3.4%), and Hiaobo, with 76 cases (1.7%). Other TB diagnostic and care centers of the capital town accounted for less than 50 cases, including HIAA, PCL, CTA Nkembo, Nvolane, CS Nzeng-Ayong, etc. (Figure 2).

Furthermore, given that approximately 98% of TB cases reported in Libreville in 2022 were diagnosed at two health settings that organized TB services, namely Nkembo and CHUL, only those data (cases) were considered and analyzed using QGIS software. The latter was used to produce the geographical distribution of TB and MDR-TB cases by residential area (county) of the patients.

### 3.3. Spatial Distribution of TB Cases Diagnosed in 2022 in Libreville

Spatial mapping of incident cases by county of residence showed that a large majority of the TB cases diagnosed at the CHUL care center in 2022 were from Nzeng-Ayong county (range: 36–50 cases) and Owendo (range: 26–35 cases) (Figure 3); other TB cases originated from counties that accounted for less than 26 cases of patients who visited the CHUL care center, such as Bikele, Bissegue-Pk8, Akebe Ville, Akebe Baraka, Ozangue, Sotega, Plaine-Orety, Charbonnages, Angondje, Angondje village, etc.

When considering TB cases diagnosed at the Nkembo care center, there was a significantly higher number of TB patients from Nzeng-Ayong county (range: 356–455 cases) and Owendo (range: 256–355 cases), followed by Nkembo county, Akebe ville, Akebe Baraka, Akebe Plaine/plateau, Angondje, Angondje village, Charbonnages, Bikele, Pk11, Pk12, Pk9, Mindoube I, Mindoube II (66–255 cases), and Sotega, and Nkok (46–65 cases). Other counties accounted for less than 45 new TB cases (Figure 4 and Figure 5).

### 3.4. Distribution of Multi-Drug-Resistant Tuberculosis Cases in Libreville in 2022

Of all 66 newly diagnosed MDR-TB cases nationwide in 2022, Libreville accounted for 62.1% (41 cases) of them, for an annual incidence of 26.7 per 100,000. The mapping of all cases from Libreville showed that a relatively higher number of MDR-TB cases were from Pk9 county, which accounted for 14.6% (6 cases) of the cases, followed by Owendo, at 9.7% (4 cases). The other counties of Libreville accounted for less than four MDR-TB cases, including Nzeng-Ayong and Charbonnages, as shown in Figure 6.

## 4. Discussion

This work searched to identify high-disease-burden areas and perform the spatial mapping of TB and MDR-TB cases diagnosed in 2022 in Libreville, Republic of Gabon, to enhance national strategies for TB control. It was observed that the capital city of Libreville had the highest number of new TB cases compared to other health regions, and most of them were diagnosed at the Nkembo and CHUL care centers. Geographically, new TB cases were highly concentrated in the Nzeng-Ayong and Owendo counties. This work also showed that the majority of TB-MDR cases were from Pk9 county and Owendo. The emergence of MDR-TB hampers the efforts to control TB [6], particularly in TB-endemic countries such as those in the WHO central Africa subregion, including the Republic of Gabon. Obviously, although the number of MDR-TB cases diagnosed in 2022 seems small, it is important to consider that even one single case is too much, given the fact that it represents a real challenge for healthcare providers and a considerable health risk for the exposed members of the community.

The Republic of Gabon is known as one of the high-burden countries not only for TB but also for MDR-TB, which has triggered efforts to establish both molecular surveillance and clinical care capacity nationwide in recent years [9,10,11]. In Gabon, resistance to Rifampicin has been estimated to be 33% among newly infected patients [12]. Other previous studies have shown that resistance to fluoroquinolone medication is due to gyrase (gyrA) mutations in this central African country. The presence of gyrA mutations in a TB patient helps predict not only the presence, but also the level of fluoroquinolone resistance, which allows care givers to adapt the drug regimen to the *M. tuberculosis* strain that causes individual infection [13].

Our study showed that Libreville had the highest MDR-TB incidence in 2022. The distribution of new MDR-TB cases within Libreville in 2022 according to area of residence showed that most cases were from Pk9, followed by Owendo, Nzeng-Ayong, and Charbonnages counties. A previous nationwide retrospective study that analyzed aggregate TB data (2014–2021) in Gabon showed high prevalence of MDR-TB, and most patients were young (25–35 years) [14]. Another study by Assiana and colleagues (2020) [15] showed a relatively high MDR-TB prevalence of 379 cases per 100,000 in the Gabonese population.

The present study, which was conducted in Libreville, shows the limited decentralization of TB programs, since 98% of patients were given care at two health facilities. It was mentioned above that Libreville had only four TB diagnostic and care centers for nearly one million inhabitants. This means that approximately 250,000 individuals were covered by one healthcare setting, while the WHO recommends 50,000–100,000 per health facility. The small number of TB care facilities and the hospital-centric health system in Gabon are likely to be the factors explaining why the patients residing in Nzeng Ayong, Owendo, and PK9 had to visit the Nkembo and CHUL care centers to seek out TB care services. This issue, noticed in the program review report of 2020, is being well addressed by the NTP, as the tuberculosis services were expanded from four to twelve between 2020 and 2022.

Regarding the health, social, and lifestyle-related determinants of TB in Gabon, the WHO Global report identified four major factors, including HIV/AIDS, alcohol use disorders, undernutrition, and diabetes in total of 6310 new TB cases in 2022. From them, HIV accounted for more than half of the cases. In poor counties with high population density, such as Nzeng Ayong, Owendo, and PK9, undernutrition, HIV, and unsafe alcohol use are common. This might explain the reason why TB is the most prevalent in those counties. A similar TB burden study was reported in Cameroon in 2022, particularly in the cities of Yaoundé and Douala, with Nkomkana, Awae, and Odza in Yaounde and Newbell and Boko in Douala as high-burden counties [16].

This study highlights the need to integrate the spatial surveillance in the tuberculosis program implementation. Clearly, this study has identified high-burden counties within Libreville; thus, it may help the NTP be more efficient in resource allocation to handle the TB burden. Nonetheless, it is limited by its retrospective design regarding data collection. It is preferred to obtain the spatial data in a progressive manner. This can help implement specific interventions simultaneously.

Suggested directions for future research include delving deeper into the public health implications of these findings. This involves analysis from socioeconomic, environmental, population density, and healthcare accessibility perspectives.

## 5. Conclusions

This work consisted of mapping TB and MDR-TB cases in Libreville, providing valuable insights into the spatial distribution of the disease. It revealed that Nzeng-Ayong and Owendo counties were the high-TB-morbidity-burden areas, whereas Pk9 was the high-burden county for MDR-TB. These findings call for the PNLT to prioritize public health interventions in these areas to improve case detection, treatment adherence, and ultimately reduce the TB and MDR-TB burden in Libreville. Further investigations are indispensable to determine possible determinants that might elucidate the increased TB morbidity in Libreville.

## Figures and Tables

**Figure 1 tropicalmed-11-00008-f001:**
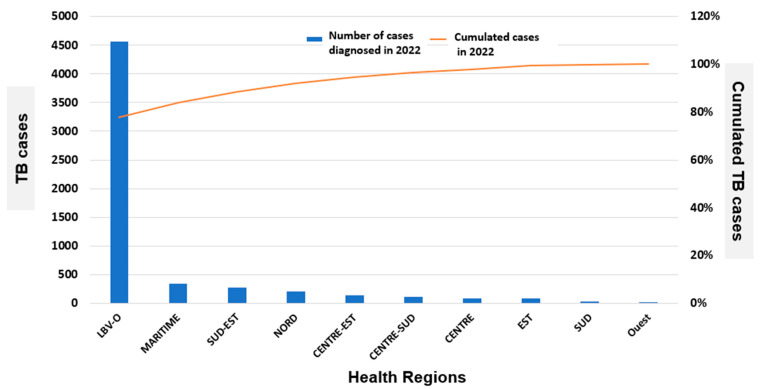
Nationwide distribution of tuberculosis (TB) in Gabon in the year 2022 by health region. Legend: LBV-O, Libreville–Owendo; TB, tuberculosis.

**Figure 2 tropicalmed-11-00008-f002:**
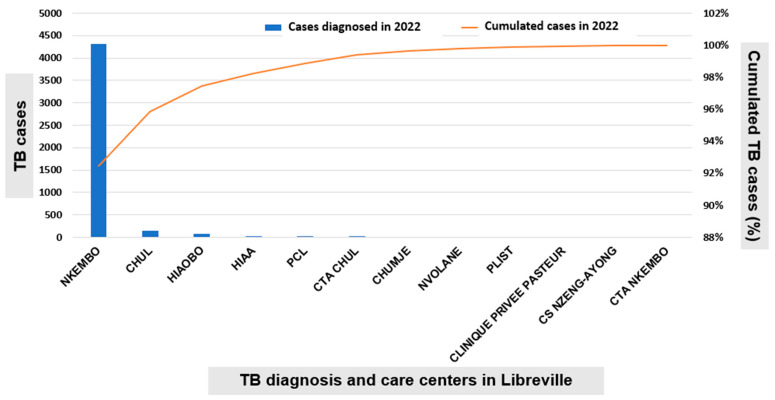
Distribution of TB cases in Libreville in the year 2022 by treatment center. Legend: CHUL, University Hospital of Libreville. CTA: Centre de Traitement Ambulatoire. CS: Centre de Santé. PCL: polyclininque de Libreville.

**Figure 3 tropicalmed-11-00008-f003:**
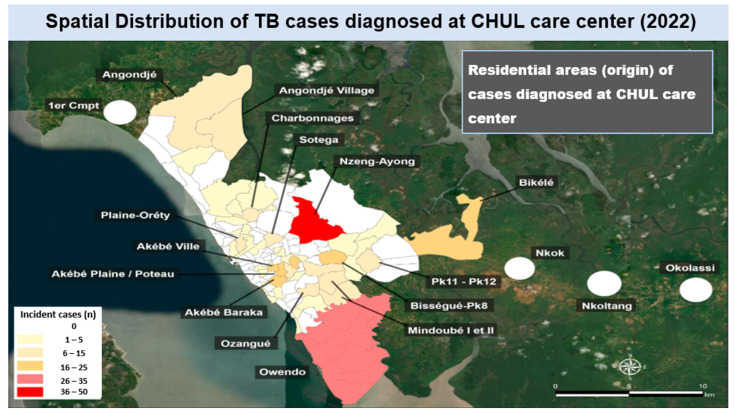
Spatial distribution of tuberculosis (TB) cases diagnosed at the CHUL care center in the year 2022 by county of residence.

**Figure 4 tropicalmed-11-00008-f004:**
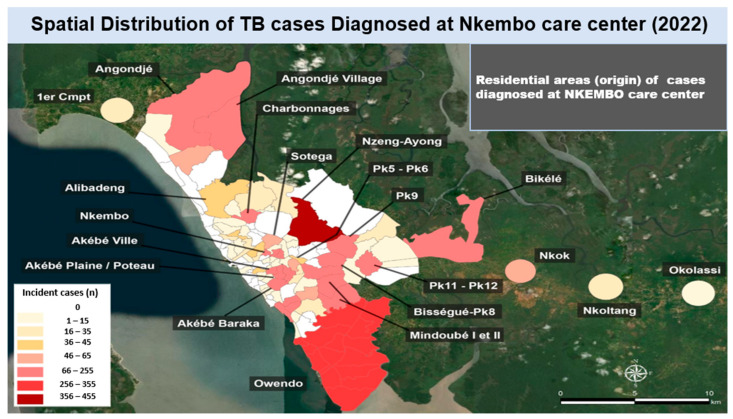
Spatial distribution of tuberculosis (TB) cases diagnosed at the Nkembo care center in the year 2022 by county of residence.

**Figure 5 tropicalmed-11-00008-f005:**
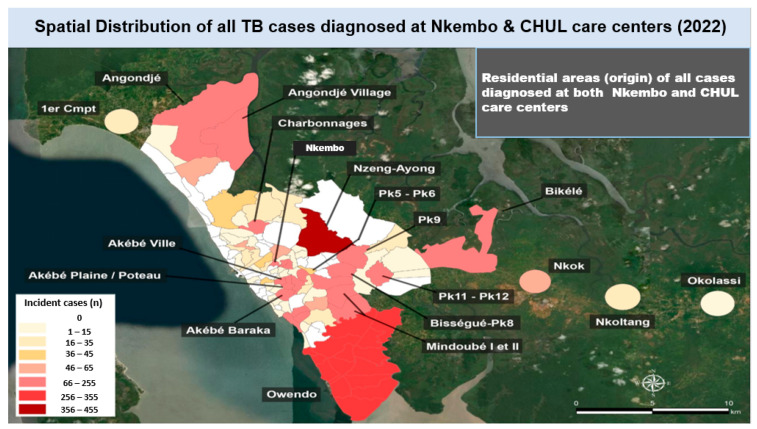
Spatial distribution of all tuberculosis (TB) cases diagnosed at the CHUL and Nkembo care centers in the year 2022 by county of residence.

**Figure 6 tropicalmed-11-00008-f006:**
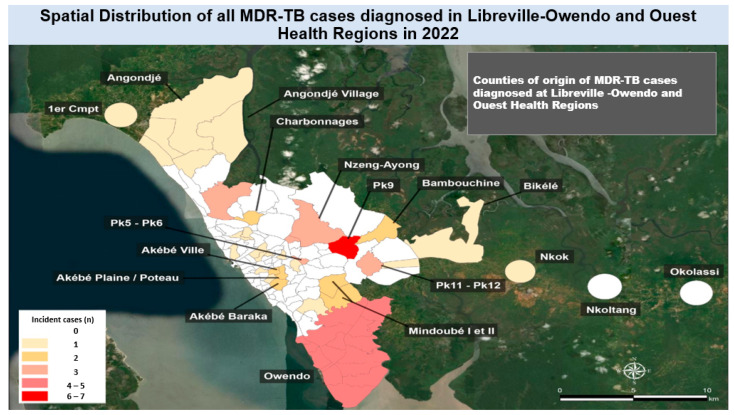
Spatial distribution of multi-drug-resistant tuberculosis (MDR-TB) cases diagnosed in Libreville in the year 2022 by county of residence.

**Table 1 tropicalmed-11-00008-t001:** Tuberculosis case distribution by sex, CDT Nkembo and CHUL, Libreville 2022.

	Male	Female	Total
CDT Nkembo	3232	1077	4309
CDT CHUL	132	25	157
SUM	3364	1102	4466

**Table 2 tropicalmed-11-00008-t002:** Tuberculosis case distribution by age group, CDT Nkembo and CHUL, Libreville 2022.

	<15 Year	15–49 Years	50+ Years
Age group	253	3537	676
%	6%	79%	15%

CDT: Centre de Diagnostic et de Traitement; CHUL: Centre Hospitalo-Universitaire de Libreville.

**Table 3 tropicalmed-11-00008-t003:** Case distribution by tuberculosis type, CDT Nkembo and CHUL, Libreville 2022.

	Bacteriologically Confirmed	Clinically Confirmed	Extrapulmonary TB
Number	3266	764	436
%	73%	17%	10%

## Data Availability

Data collected belong to the Gabonese National Tuberculosis Program, and related information can be obtained upon request to the authors (F.L., C.M. and N.R.N.).

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
