# Peer review of "Incidence and Spatial Mapping of Tuberculosis and Multidrug-Resistant Tuberculosis in Libreville, Republic of Gabon, in 2022"

_tropicalmed, 2025, doi:10.3390/tropicalmed11010008_

Round 1

Reviewer 1 Report

Comments and Suggestions for Authors

Comments for the Authors

TB mapping is a valuable tool for identifying TB clusters and high-risk areas, which aids in effective disease monitoring and control. Overall, it's an interesting work and will certainly help in monitoring and controlling the disease effectively. Here are some suggestions for the author.

  • In the background of the abstract, add 1-2 more lines.
  • The discussion part is not normally included in the abstract. Check with the journal policy and instead include the conclusion part in the abstract.
  • From lines 71 to 82, instead of stating the findings of one study in a whole paragraph, cite some other research studies conducted in Gabon and other African countries and only highlight the main findings related to the disease.
  • From lines 83 to 104, there are four paragraphs discussing almost the same or related topics. Rewrite or rearrange these paragraphs into two paragraphs, and also avoid repetition of the sentences.
  • Make separate headings for data collection, data analysis, and ethical considerations in the methods part and add relevant information into each individual heading. Also, include a heading on sampling.
  • No data is presented on the MDR-TB cases in the results section, and remapping is also not available.
  • Limited data is presented in the results section. No information or data is available on the patients and their demographics, or other parameters of the disease.
  • No information on presented on the severity of the disease and the patient's condition, and also on morbidity and mortality.
  • Add more relevant references to the manuscript. Conduct a comprehensive literature review to cite references in the introduction and discussion parts.
Comments on the Quality of English Language

The quality of English can be improved further.

Author Response

TB mapping is a valuable tool for identifying TB clusters and high-risk areas, which aids in effective disease monitoring and control. Overall, it's an interesting work and will certainly help in monitoring and controlling the disease effectively. Here are some suggestions for the author.

  1. In the background of the abstract, add 1-2 more lines.
    • Answer: thank you; we added a few lines and corrected sentences in the Abstract. See lines 18-23
  1. The discussion part is not normally included in the abstract. Check with the journal policy and instead include the conclusion part in the abstract.
  • Answer: Thanks for the suggestion; we include the main part of Conclusion section in the revise version of the Abstract.
  1. From lines 71 to 82, instead of stating the findings of one study in a whole paragraph, cite some other research studies conducted in Gabon and other African countries and only highlight the main findings related to the disease.
  • Answer: Thanks for the suggestion. We added more citations related to studies conducted in countries of the central Africa region.
  1. From lines 83 to 104, there are four paragraphs discussing almost the same or related topics. Rewrite or rearrange these paragraphs into two paragraphs, and also avoid repetition of the sentences.
  • Answer: we rewrite the Introduction section and kept only sentences that are in relation with the study topic, thus reducing the number of paragraphs. Lines 49-86
  1. Make separate headings for data collection, data analysis, and ethical considerations in the methods part and add relevant information into each individual heading. Also, include a heading on sampling.
  • Answer: Thanks for this suggestion. We used headings and subheadings in the Methods section of the revised manuscript. 90-168
  1. No data is presented on the MDR-TB cases in the results section, and remapping is also not available.
  • Answer: Thank you. Information on MDR-TB is provided in the Results section, Lines 229 – 238 of the revised manuscript.
  1. Limited data is presented in the results section. No information or data is available on the patients and their demographics, or other parameters of the disease.
  • Answer: Thank you for this point. This point is now developed in the revised version. Lines 171-180
  1. No information on presented on the severity of the disease and the patient's condition, and also on morbidity and mortality.
  • Answer: Thank you for this concern. The study objective was to determine TB and MDR-TB incidence, and provide their geographical distribution within Libreville to inform evidence-based public health intervention designed to reduce the disease burden. TB severity and treatment outcomes are part of another study.
  1. Add more relevant references to the manuscript. Conduct a comprehensive literature review to cite references in the introduction and discussion parts.
  • Answer: we added relevant references in the Introduction and Discussion sections as you suggested.
  1. The quality of English can be improved further.
    • Answer: we had the revised manuscript proofread by a Medical English editor. We think that the quality of the paper has improved.

Reviewer 2 Report

Comments and Suggestions for Authors

This is a really important and timely study on the public health challenge of Tuberculosis in Libreville, Gabon. Using GIS mapping to pinpoint hotspots for TB and drug-resistant TB is a powerful approach, and the data you've gathered could be incredibly valuable for strengthening the national TB program.

While the core idea is strong, the manuscript would benefit from addressing some key concerns around the methodology, analysis, and clarity of reporting to fully meet its potential.

Major Points to Address:

  1. Study Design & Methodological Clarity
  • You focus on just two health centers (Nkembo and CHUL), which, while covering most cases, leaves out smaller centers. Could you please explain the reasoning for this selection in more detail? It would be helpful to discuss how excluding these smaller centers might have skewed the spatial patterns you observed, and if you did any checks (like a sensitivity analysis) to account for this.
  • The spatial resolution isn't entirely clear. It seems like you might be working with aggregated counts per area rather than precise patient locations. Using broad case ranges (e.g., 356–455 cases) limits the map's precision. Please clarify the level of detail you had for the locations and why you chose to group the data into these categories.
  1. Incidence Calculations
  • When you present incidence rates (like 509 per 100,000), it's not clear which population numbers you used as the denominator for each neighborhood. Please specify the source and the year of the population data, and briefly explain how you calculated the incidence for each specific area on the maps.
  1. Data Quality & Completeness
  • Since the study relies on medical records, it's important to give the reader a sense of the data's quality. A note on how complete the records were, how you handled any missing address information, or the potential for misclassifying a patient's home area would be very useful.
  • The point made in the introduction that only a third of TB patients were tested for HIV is concerning. This should be acknowledged as a significant limitation when discussing the reliability of any associated risk factors.
  1. Spatial Analysis & GIS Methods
  • The description of how you did the mapping is a bit sparse. To help others understand and replicate your work, please add details like:
    • The specific projection or coordinate system used.
    • Which GIS software and map layers you worked with.
    • The exact type of analysis performed (e.g., was it simple choropleth mapping, kernel density estimation, or something else?).
    • Whether you used any statistical tests (like Moran's I or SaTScan) to formally identify clusters.
  • Currently, the maps seem more descriptive than analytical. Without a formal statistical test to back it up, using the term "hotspot" might be overstating the findings. I'd recommend either applying a proper cluster detection analysis or toning down the language (e.g., "areas of high case density").
  1. Interpretation of Results
  • In the discussion, you often link high TB rates to factors like poverty and overcrowding. However, your study didn't actually collect data on these factors. It's best to avoid making these direct causal links and instead frame them as plausible explanations, supported by references to other literature.
  • The comparisons with studies from Cameroon and DRC are interesting, but since those studies used different methods, the connection to your own findings in Libreville isn't always clear. Try to weave these comparisons in more carefully, focusing on what the similarities or differences might mean.
  1. MDR-TB Analysis
  • With only 41 MDR-TB cases in total, the numbers for any single area are very small (e.g., 6 cases in Pk9). It's important to highlight the uncertainty here and avoid making strong claims about "hotspots" based on such low counts.
  1. Figures & Data Presentation
  • The maps in Figures 3 through 6 need clearer legends and color scales. As they are, it's difficult to distinguish the different intensity levels. Also, remember to include the population denominators used for the incidence maps.
  • It would be much easier for the reader to see the big picture if you could provide a single, comprehensive map that layers both the TB and MDR-TB data. At the very least, ensuring consistency between the different maps would be a big improvement.

Minor Points:

  • The introduction has some repetition of well-known global TB statistics; this section could be tightened up.
  • There are a few typos throughout (e.g., "hotpots," "commmun," inconsistent spacing) that a thorough proofread should catch.
  • Stating "Ethical approval: Not applicable" requires a brief justification. Even with anonymized data, it's good practice to mention that the study adhered to relevant national regulations or institutional guidelines.
  • Some references appear to be duplicates (e.g., reference 10 and 14); please double-check the list for accuracy and formatting.

Overall

This work has the potential to make a valuable contribution to TB control in Gabon. I am recommending a major revision to strengthen the methodological foundation, add rigor to the spatial analysis, and improve the clarity of the presentation. With these changes, I believe the manuscript will be much stronger and suitable for publication.

Author Response

This is a really important and timely study on the public health challenge of Tuberculosis in Libreville, Gabon. Using GIS mapping to pinpoint hotspots for TB and drug-resistant TB is a powerful approach, and the data you've gathered could be incredibly valuable for strengthening the national TB program.

While the core idea is strong, the manuscript would benefit from addressing some key concerns around the methodology, analysis, and clarity of reporting to fully meet its potential.

Major Points to Address:

  1. Study Design & Methodological Clarity
  • You focus on just two health centers (Nkembo and CHUL), which, while covering most cases, leaves out smaller centers. Could you please explain the reasoning for this selection in more detail? It would be helpful to discuss how excluding these smaller centers might have skewed the spatial patterns you observed, and if you did any checks (like a sensitivity analysis) to account for this.
  • Answer: Thanks. We provide the reason why his study focused on Nkembo and CHUL healthcare settings in the Methods section. Briefly, almost all new cases were diagnosed there. Lines 110-116
  • The spatial resolution isn't entirely clear. It seems like you might be working with aggregated counts per area rather than precise patient locations. Using broad case ranges (e.g., 356–455 cases) limits the map's precision. Please clarify the level of detail you had for the locations and why you chose to group the data into these categories.
  • Answer: Thank you. Within each province, there are Health region which comprise a number of counties (communes), according to the Gabonese health system organization. By determining the geographical distribution of cases according to residential area, it facilitates the understanding of care providers and policymakers, and the planification of public health intervention at health area as well. The use of cases range instead of exact number of cases is due to the fact that the research team searched to visualize the density or concentration of TB/MDR-TB cases by area.
  1. Incidence Calculations
  • When you present incidence rates (like 509 per 100,000), it's not clear which population numbers you used as the denominator for each neighborhood. Please specify the source and the year of the population data, and briefly explain how you calculated the incidence for each specific area on the maps.
  • Answer: Thank you very much for this question. For each case, residential address was used to match it with the county of origin. The total number of cases was used as the numerator, whereas the county level population was used as the denominator. The county population was retrieved from demographic report from WHO Gabon, 2022.
  1. Data Quality & Completeness
  • Since the study relies on medical records, it's important to give the reader a sense of the data's quality. A note on how complete the records were, how you handled any missing address information, or the potential for misclassifying a patient's home area would be very useful.
  • The point made in the introduction that only a third of TB patients were tested for HIV is concerning. This should be acknowledged as a significant limitation when discussing the reliability of any associated risk factors.
  • Answer: Thanks for pointing out this important fact. It’s true that HIV infection plays an important role in TB outcomes among individuals infected with M. tuberculosis, and it is necessary to assess the TB – HIV coinfection. Nonetheless, as mentioned above, that was not the object of the present study. It is taken into account in another study.
  1. Spatial Analysis & GIS Methods
  • The description of how you did the mapping is a bit sparse. To help others understand and replicate your work, please add details like:
    • The specific projection or coordinate system used.
    • Which GIS software and map layers you worked with.
    • The exact type of analysis performed (e.g., was it simple choropleth mapping, kernel density estimation, or something else?).
    • Whether you used any statistical tests (like Moran's I or SaTScan) to formally identify clusters.

Answer: Thank you for raising this important point. Additional lines are added in the revised version to give more insights. The ARCGIS 10.7 software was used, and the data are those of the neighborhood boundaries of Libreville (Source: Ministry of the Interior, Format: shapefile). For neighborhoods that do not yet have recognized boundaries, a buffer zone was used to locate them. The color gradient, for its part, differentiates each neighborhood according to the number of cases observed. Lines 145-149

  • Currently, the maps seem more descriptive than analytical. Without a formal statistical test to back it up, using the term "hotspot" might be overstating the findings. I'd recommend either applying a proper cluster detection analysis or toning down the language (e.g., "areas of high case density").
  • Answer: Thank you for this important comment. In thew revised version of the manuscript, we use the term “high burden area” instead of “hotspot”; and “case density” is also used, though scarcely.
  1. Interpretation of Results
  • In the discussion, you often link high TB rates to factors like poverty and overcrowding. However, your study didn't actually collect data on these factors. It's best to avoid making these direct causal links and instead frame them as plausible explanations, supported by references to other literature.
  • The comparisons with studies from Cameroon and DRC are interesting, but since those studies used different methods, the connection to your own findings in Libreville isn't always clear. Try to weave these comparisons in more carefully, focusing on what the similarities or differences might mean.
  • Answers: Yes, you are right; the methodology used in studies carried out in Cameroon and DRC is different from the one we used. We just point ou similarities in terms of study outcomes. Our study did not analyze the relationship between TB burden and potential risk factors. The information given on risk factors are from previous studies; it is used in our report as background information.

  1. MDR-TB Analysis
  • With only 41 MDR-TB cases in total, the numbers for any single area are very small (e.g., 6 cases in Pk9). It's important to highlight the uncertainty here and avoid making strong claims about "hotspots" based on such low counts.
  • Answer: we understand the point; but even 1 new MDR-TB diagnosed is already a problem, because it can be a source of spread of drug resistant strain which may render the disease difficult to control. Thus, having 41 new resistant cases in a 12-month period is too much and should be taken seriously by care providers and health policymakers.
  1. Figures & Data Presentation
  • The maps in Figures 3 through 6 need clearer legends and color scales. As they are, it's difficult to distinguish the different intensity levels. Also, remember to include the population denominators used for the incidence maps.
  • It would be much easier for the reader to see the big picture if you could provide a single, comprehensive map that layers both the TB and MDR-TB data. At the very least, ensuring consistency between the different maps would be a big improvement.
  • Answer: Thank you for this point. Additive explanation is given in the methodology.

Minor Points:

  • The introduction has some repetition of well-known global TB statistics; this section could be tightened up.
  • There are a few typos throughout (e.g., "hotpots," "commmun," inconsistent spacing) that a thorough proofread should catch.
  • Answer: we corrected those typos; the term hotspot is replaced by high burden area in the revised version of the manuscript.
  • Stating "Ethical approval: Not applicable" requires a brief justification. Even with anonymized data, it's good practice to mention that the study adhered to relevant national regulations or institutional guidelines.
  • Answer: we added justification as suggested by the reviewer.
  • Some references appear to be duplicates (e.g., reference 10 and 14); please double-check the list for accuracy and formatting.
  • Answer: Thank you for this concern. The revised version is now well referenced.

Round 2

Reviewer 1 Report

Comments and Suggestions for Authors

The authors have addressed the major concerns and revised the manuscript in line with the reviewers’ suggestions. However, the Results section still requires some improvement. It would strengthen the paper if the authors add a table summarizing patient demographics and other relevant clinical parameters.

In addition, the Introduction has become very brief. Important scientific literature highlighting the burden of tuberculosis in the country and across the African region has been removed. The authors are encouraged to reintroduce and cite relevant studies in the Introduction to better contextualize the problem.

Author Response

The authors have addressed the major concerns and revised the manuscript in line with the reviewers’ suggestions. However, the Results section still requires some improvement. It would strengthen the paper if the authors add a table summarizing patient demographics and other relevant clinical parameters.

Answer: Thank you for support to strengthen the paper. Some tables are added in the result section from 174 to 190 lines.

In addition, the Introduction has become very brief. Important scientific literature highlighting the burden of tuberculosis in the country and across the African region has been removed. The authors are encouraged to reintroduce and cite relevant studies in the Introduction to better contextualize the problem.

Answer: Thank you for raising this point. Additive studies are added in the introduction part to better contextualize the problem. See lines 57-61.

Reviewer 2 Report

Comments and Suggestions for Authors

I would like to thank the authors for their thorough, careful, and constructive responses to all reviewer comments. The revised manuscript shows a clear and consistent improvement in methodological transparency, clarity of reporting, and appropriateness of interpretation.

The authors have satisfactorily addressed the major methodological concerns by:

  • Clearly justifying the focus on the Nkembo and CHUL health centers and integrating this rationale into the Methods section;

  • Clarifying the spatial resolution and explaining the use of case ranges as a visualization strategy to reflect case density rather than exact counts;

  • Providing explicit information on population denominators and the sources used for incidence calculations;

  • Expanding the description of the GIS methodology, including software, spatial layers, and mapping procedures;

  • Appropriately revising the terminology from “hotspots” to “high burden areas” to better reflect the descriptive nature of the spatial analysis;

  • Correcting typographical issues, improving figure explanations, and strengthening ethical justifications.

Importantly, the discussion has been refined to avoid unsupported causal inferences, with socioeconomic and contextual factors now framed appropriately as background information supported by existing literature rather than direct findings of this study.

Overall, the manuscript is now clear, coherent, and scientifically sound. It provides valuable and policy-relevant insights into the spatial distribution of TB and MDR-TB in Libreville and represents a meaningful contribution to the field of spatial epidemiology and tuberculosis control in Central Africa.

I am satisfied with the revisions and recommend the manuscript for publication in its current form.

Author Response

I would like to thank the authors for their thorough, careful, and constructive responses to all reviewer comments. The revised manuscript shows a clear and consistent improvement in methodological transparency, clarity of reporting, and appropriateness of interpretation.

The authors have satisfactorily addressed the major methodological concerns by:

  • Clearly justifying the focus on the Nkembo and CHUL health centers and integrating this rationale into the Methods section;
  • Clarifying the spatial resolution and explaining the use of case ranges as a visualization strategy to reflect case density rather than exact counts;
  • Providing explicit information on population denominators and the sources used for incidence calculations;
  • Expanding the description of the GIS methodology, including software, spatial layers, and mapping procedures;
  • Appropriately revising the terminology from “hotspots” to “high burden areas” to better reflect the descriptive nature of the spatial analysis;
  • Correcting typographical issues, improving figure explanations, and strengthening ethical justifications.

Importantly, the discussion has been refined to avoid unsupported causal inferences, with socioeconomic and contextual factors now framed appropriately as background information supported by existing literature rather than direct findings of this study.

Overall, the manuscript is now clear, coherent, and scientifically sound. It provides valuable and policy-relevant insights into the spatial distribution of TB and MDR-TB in Libreville and represents a meaningful contribution to the field of spatial epidemiology and tuberculosis control in Central Africa.

I am satisfied with the revisions and recommend the manuscript for publication in its current form.

Answer: Thank you for all your constructive feedback. Appreciated.
